# Prediction Power on Cardiovascular Disease of Neuroimmune Guidance Cues Expression by Peripheral Blood Monocytes Determined by Machine-Learning Methods

**DOI:** 10.3390/ijms21176364

**Published:** 2020-09-02

**Authors:** Huayu Zhang, Edwin O. W. Bredewold, Dianne Vreeken, Jacques. M. G. J. Duijs, Hetty C. de Boer, Adriaan O. Kraaijeveld, J. Wouter Jukema, Nico H. Pijls, Johannes Waltenberger, Erik A.L. Biessen, Eric P. van der Veer, Anton Jan van Zonneveld, Janine M. van Gils

**Affiliations:** 1Einthoven Laboratory for Vascular and Regenerative Medicine, Department of Internal Medicine, Leiden University Medical Center, Albinusdreef, 22333 ZA Leiden, The Netherlands; h.zhang@lumc.nl (H.Z.); o.w.bredewold@lumc.nl (E.O.W.B.); d.vreeken@lumc.nl (D.V.); j.m.g.j.duijs@lumc.nl (J.M.G.J.D.); h.c.de_boer@lumc.nl (H.C.d.B.); e.p.van_der_veer@lumc.nl (E.P.v.d.V.); a.j.van_zonnveveld@lumc.nl (A.J.v.Z.); 2Department of Cardiology, University Medical Center Utrecht, Heidelberglaan, 1003584 CX Utrecht, The Netherlands; a.o.kraaijeveld-3@umcutrecht.nl; 3Department of Cardiology, Leiden University Medical Center, Albinusdreef, 22333 ZA Leiden, The Netherlands; j.w.jukema@lumc.nl; 4Department of Cardiology, Catharina Hospital, Michelangelolaan, 25623 EJ Eindhoven, The Netherlands; n.h.j.pijls@tue.nl; 5Department of Cardiology, Maastricht University Medical Center, P. Debyelaan, 256202 AZ Maastricht, The Netherlands; johannes.waltenberger@ukmuenster.de; 6Department of Pathology, Cardiovascular Research Institute Maastricht (CARIM), University of Maastricht, Universiteitssingel, 506229 ER Maastricht, The Netherlands; erik.biessen@mumc.nl

**Keywords:** cardiovascular diseases, monocytes, machine-learning methods, neuroimmune guidance cues

## Abstract

Atherosclerosis is the underlying pathology in a major part of cardiovascular disease, the leading cause of mortality in developed countries. The infiltration of monocytes into the vessel walls of large arteries is a key denominator of atherogenesis, making monocytes accountable for the development of atherosclerosis. With the development of high-throughput transcriptome profiling platforms and cytometric methods for circulating cells, it is now feasible to study in-depth the predicted functional change of circulating monocytes reflected by changes of gene expression in certain pathways and correlate the changes to disease outcome. Neuroimmune guidance cues comprise a group of circulating- and cell membrane-associated signaling proteins that are progressively involved in monocyte functions. Here, we employed the CIRCULATING CELLS study cohort to classify cardiovascular disease patients and healthy individuals in relation to their expression of neuroimmune guidance cues in circulating monocytes. To cope with the complexity of human datasets featured by noisy data, nonlinearity and multidimensionality, we assessed various machine-learning methods. Of these, the linear discriminant analysis, Naïve Bayesian model and stochastic gradient boost model yielded perfect or near-perfect sensibility and specificity and revealed that expression levels of the neuroimmune guidance cues SEMA6B, SEMA6D and EPHA2 in circulating monocytes were of predictive values for cardiovascular disease outcome.

## 1. Introduction

Cardiovascular diseases (CVD) remain a leading cause of death in the more economically developed countries, despite improvements in surgical and drug treatments. Much of the CVD-related mortality and morbidity is attributable to atherosclerosis [1]. Atherosclerosis is a systemic chronic inflammatory and immune disease [2,3]. Monocytes and their derived macrophages play a key role in the development of atherosclerosis. Under conditions of dyslipidemia and chronic systemic inflammation, circulating monocytes and the endothelium become activated, resulting in monocyte infiltration and differentiation into macrophages in the vessel wall. Upon the excessive uptake of lipids, these macrophages become foam cells and participate decisively in the development and exacerbation of atherosclerosis, coronary stenosis and its clinical sequela, such as acute myocardial infarctions [3,4,5,6,7].

Neuroimmune guidance cues (NGCs) comprise the netrin, semaphorin, ephrin and slit families of proteins of ligands and receptors, which were originally characterized to direct cell and axon migration during neural development. In the last two decades, it has become increasingly clear that these proteins can also play a major role in (pathological) immune responses by directly regulating leukocyte trafficking and directly impacting the pathogenesis of atherosclerosis [8,9,10]. Indeed, numerous studies using murine atherosclerosis models have found multifaceted roles of NGCs in the development of atherosclerosis [11,12,13,14,15]. In addition, several observations also support a role for NGCs in human CVD. For instance, three NGC genes are located on human chromosome 1, in the locus that has been identified as the premature myocardial infarction susceptibility locus [16]. In addition, the axonal guidance pathway is found enriched with genetic variants that have significant associations with CVD, and several novel genetic risk loci for CVD contain NGCs genes [17,18]. However, whether the monocytic expression of NGCs is informative for human CVD has not been described yet.

Transcriptomics can reveal key alterations in biological processes causing human diseases, thereby present novel instruments that are not only useful for the understanding of the disease mechanisms but, also, for molecular diagnosis and clinical therapy [19]. Since monocytes are among the culprit cells of atherosclerosis development, monocytic expression levels of NGCs could provide insights into the underlying mechanisms in atherosclerosis development and can be used to improve the evidence-based treatment of CVD to reduce the global burden of this disease.

The CIRCULATING CELLS study was designed to study the role of several cellular mediators of atherosclerosis as biomarkers of CVD to predict the susceptibility of patients to the progression of CVD [20]. By applying different machine-learning methods (also known as predictive modeling methods) on the gene expression data of peripheral monocytes from the CIRCULATING CELLS study cohort, we investigated whether monocytic NGC expression is informative to distinguish between healthy individuals and CVD patients. As machine-learning methods are developed to explore complex relationships between predictors and outcomes, they are suitable tools to tackle the difficulties due to the complexity of human datasets featured by noisy data, nonlinearity and multidimensionality. Some machine-learning methods take simplistic approaches and work with linear relationships between features and outcomes, while other methods are more complex and are able to capture nonlinear patterns and to tolerate a low information-noise ratio, owing to the difference of their pre-assumptions and learning logics. We compared the performance of multiple modeling methods to explore the best predicting potential of our dataset. In our study, we included commonly used models like logistic regression and linear discriminant analysis, as well as more complex nonlinear models and tree-based models. Linear models, like partial least square, have supervised dimension reduction functionality, which benefits model performances in the case of high between-feature correlations. Nonlinear models, like Naïve Bayesian, make probabilistic calls based on the information provided by the features independently, possibly performing better in situations when between-feature correlations are low. In addition to modeling of the data itself, stochastic gradient boost also models the residuals, thereby increasing the learning ability when the information-noise ratio is low. We compared the results of the different modeling methods to gain insights on the nature of the dataset. Altogether, this allowed us to give a proof of concept that the expression of a small set of functional genes can be a prediction value for complex diseases like CVD.

## 2. Results

### 2.1. NGC Expressions in Monocytes and Feature Selections

We made use of the CIRCULATING CELLS study cohort [20] to address the question of whether NGC expression profiles of circulating blood cells can be related to cardiovascular health. From 368 subjects out of this cohort (CVD patients and healthy controls), the transcriptomes of their CD14-positive monocytes were profiled (Figure 1). The individuals received different treatments and medications, and some of them suffered from other diseases, resembling the reality of complexity of most human cohorts (Table 1). Next, we aimed to classify CVD patients and healthy individuals using the differential expression of the NGC transcripts

To that end, we first sought to include NGCs with high expression levels and good univariate correlation with the outcome. Figure 2a shows the expression of NGCs in the cohort. Based on the detection threshold of the profiling platform, NGCs with signals higher than 6.75 (log_2_ scale unless specified otherwise) were unconditionally included in the modeling as potential features. To validate the microarray analysis, we compared the monocytic NGC expression profile obtained by microarray to that obtained by real-time PCR. Both methods showed a similar expression profile, with the exception of SEMA3E (Appendix A). To gain understanding of univariate correlation of the features to the outcome, violin plots of the NGC expressions were created to compare the distribution of NGC expression levels in both the CVD group and healthy group (Figure 2b and Appendix A). The ranges of expressions showed overlaps in both groups, suggesting that the univariate prediction power will be minimal. In addition, with the ranges being widespread, the information to noise ratio is relatively low in this dataset. To quantify the univariate correlation of NGC expressions to the disease status, we calculated the *p*-value with two mean *t*-tests between the CVD group and healthy group. A volcano plot was created to observe the *t*-test *p*-value in relation to the fold change (Figure 3a). We identified several NGC ligands and receptors to be significantly different between CVD patients and healthy individuals, although with small fold changes. Among the significantly different genes, 10 had expression signals lower than 6.75. These 10 were added to the modeling procedure despite their low expression levels. In total, 35 NGC genes were further to be used as features in subsequent modeling.

Finally, to avoid instability caused by between-feature collinearity in some models, we calculated between-feature correlations of the expressions of selected NGCs (Figure 3b). No pairwise correlations of NGC expressions exceeded the threshold of 0.75, suggesting that there would be a minimal influence of collinearity. Therefore, none of the selected NGCs were eliminated based on between-feature correlations.

### 2.2. Gender and Age are Unlikely to be Confounding Factors in the Current Study

Gender and age are conventional confounding factors in clinical situations when revealing relationships between measurements of phenotypes and diseases. For machine learning, if age or gender affect both the features and the outcome, they would be confounding factors by definition.

Firstly, we examined the relationship of age to NGC expressions and disease outcome. NGC expressions plotted against age in scatterplots with linear fittings showed no significant correlation between age and the NGC expressions (Figure 4a). The age ranges of both groups overlap, although the younger age dominates in the healthy group (Figure 4b). These observations suggested that age would add an additional prediction power in our modeling but would not be a confounding factor, as it does not link directly to NGC expressions.

Next, we examined the relationship of sex to NGC expressions and disease outcomes. Using violin plots, we compared the distribution of NGC expressions in both sexes (Figure 5a). The distribution of NGC expressions was barely affected by sex, including the X-linked PLXNA3, PLXNB3 and EFNB1 genes (Figure 5a). There were six NGCs with significantly different expressions comparing males to females, albeit with a very small fold change relative to the variations (Figure 5b). Cross-tabulations of sex and disease outcomes were made, and the frequency distribution of both genders was similar among CVD patients and healthy individuals (Figure 5c). Therefore, sex was also unlikely to be a confounding factor in this study. Regardless, sex and age were included in our modeling process, as it is common practice to control for these conventional confounding factors.

### 2.3. Performance of Different Models

Different machine-learning methods were applied, and optimal tuning was obtained for each model listed in Table 2. The performance statistics for each of the models was calculated and summarized in Figure 6. As measurements for model stability, we examined standard deviations of the cross-validations for accuracies and Cohen’s Kappa. Model performance could be categorized in four groups (Figure 6a–d). (1) Partial least squares, support vector machine and nearest shrunken centroids models failed entirely to model the informative part of the gene expression data, as was revealed by having bottom-line accuracy (not better than predicting all the individuals to be CVD) and zero Cohen’s Kappa value in the cross-validation, training set and test set. (2) Most models—namely, logistic regression, k-nearest neighbors, mixture discriminant analysis, flexible discriminant analysis, bagged CART (classification and regression tree), random forest and single C5.0 tree—suffered from overfitting, as was characterized by far better performances of models in the training set than in the test set. This can be explained by the modeling process trying to polish these models to perfectly predict the outcome based on the information in the training dataset. However, the modeling for the training set could not be generalized to the test set data. (3) K-nearest neighbors and flexible discriminant models had overall better prediction powers over random guesses but were still not up to standard. (4) Linear discriminant analysis, Naïve Bayesian and stochastic gradient boosting models performed best compared with the other models, within both the training set and test set at an accuracy of more than 0.98 and Cohen’s Kappa more than 0.75 in the test set. These results indicate that the linear discriminant analysis, Naïve Bayesian and stochastic gradient boosting models were able to translate the informative part of NGC expression data into disease outcome.

The three best-performing models all reached a sensitivity of 1 in the test set, meaning that they were able to discriminate healthy individuals from CVD patients (Figure 6c). Cohen’s kappa (*κ*) values were 0.79, 0.79 and 1 for the linear discriminant analysis, Naïve Bayesian model and stochastic gradient boost model, respectively (Figure 6b). The lower Cohen’s kappa (*κ*) for the former two models were due to the misclassification of one healthy individual as a CVD patient (Table 3). In the prediction of the training set, the linear discriminant analysis had lower sensitivity in the training set due to the misclassification of three healthy individuals to the CVD group (Table 3). Interestingly, the three misclassified healthy individuals still had higher modeled probability to be healthy than all but one misclassified CVD patient, suggesting that the sensitivity problem can be solved by an alternative cutoff value of the classification probability. When the cutoff value was altered from the original 0.5 to 0.28, the linear discriminant analysis achieved the same ideal sensitivity as the Naïve Bayesian model (Appendix A). However, to prove the external efficiency of the alternative cutoff, another independent test set would be necessary, which is not feasible considering the size of this study.

### 2.4. Features with the Most Importance in the Models

Apart from age, the most important three features determined by the linear discriminant analysis and Naïve Bayesian model were PLXNC1, DSCAM and DCC, while the most important three features determined by stochastic gradient boost were SEMA6B, SEMA6D and EPHA2 (Figure 7a–c). As we noted before, age was determined to be important contributor to the model but was hardly a confounding factor, considering the weak correlation of age with NGC expressions (Figure 7a–c). The functional relevance of the top features will be discussed in the Discussion section.

## 3. Discussion

In this study, we used NGC expressions of peripheral blood monocytes for the prediction of CVD. To reveal the true prediction power of monocytic NGC expression profiles, we performed cross-validation and validation using a pseudo-external test set with conventional confounding factors controlled. Of the models, Naïve Bayesian model and stochastic gradient model had satisfactory discrimination in both the training test and test set. The stochastic gradient model with a residual modeling mechanism was able to achieve 100% accuracy. Therefore, we have established the proof of concept that a small set of functional genes, NGCs, is of sufficient prediction power for the classification of CVD patients and healthy individuals.

There are several challenging factors in the prediction of CVDs using monocytic NGC expressions. Firstly, nonlinearity is common in the biological effect of proteins. Taking NTN1 as an example, the repellent effect of NTN1 on monocyte and macrophage migration has an optimal concentration of around 250 ng/mL; higher or lower concentrations are both less effective [3,14]. The biological effect also depends on its target, as NTN1 also has opposite roles acting on smooth muscle cells or macrophages [15]. Secondly, concomitant physiological processes in certain disease conditions can systemically cancel out the change of average gene expressions if they alter the gene expression in the opposite direction. In our case, CVD patients experience changes in the monocyte subpopulations, with increased lipopolysaccharide receptors and the low-affinity FC γ receptor-positive monocytes, referred to as intermediate monocytes [21,22]. These intermediate monocytes have been shown to predict cardiovascular events in subjects referred for elective coronary angiography [23]. At the same time, there is increased mobilization of lipopolysaccharide receptor-positive and low-affinity FC γ receptor-negative naïve classical monocytes from bone marrow, a process termed monocytosis. For any changes induced by monocyte activation/differentiation, monocytosis will cancel out the change because of the added naïve classical monocyte population. Moreover, human measurements in general can be very heterogenic by nature. Even proved biomarkers suffer from false positives and false negatives because of large variations in human measurements. In this study, we sought to tackle these problems by applying multiple modeling methods, each of which incorporates special features in the aspects of the linearity requirements or the learning mechanisms.

Machine-learning methods are different from each other in various ways, including learning mechanisms and the assumptions made on the features. For a given dataset, choosing a model with suitable learning mechanisms and proper assumptions of the features can benefit the performance of modeling. In the current dataset, the linear discriminant analysis and the Naïve Bayesian model both adopted the same features, with identical weights on each feature, but the Naïve Bayesian model had better performance (Figure 6 and Figure 7a,b). As both models are based on multivariate probability densities, the difference of the model performance should result from the different intrinsic assumptions and learning mechanisms of the models. The linear discriminant analysis assumes a multidimensional Gaussian distribution of the feature data, while the Naïve Bayesian model works with a more flexible distribution. Instead, the Naïve Bayesian model makes a strong assumption that all features are independent, so that the conditional probability of one class will be simply the product of the probability densities of all features. In addition, Naïve Bayesian could model nonlinear relationships between features and the outcome. Due to smaller numbers of individuals in the healthy group, distributions were sometimes non-normal (Figure 2b). As previously described, the features in this dataset have relatively low pairwise covariance (Figure 3b), so that the independent-feature assumption required by the Naïve Bayesian model is very likely to be acceptable. Taken together, the structure in the current dataset favors the Naïve Bayesian model, which explains its better performance. The top 10 important NGCs chosen in the linear discriminant analysis and Naïve Bayesian models were all significantly differently expressed in the two mean *t*-tests, except NTNG1 (Figure 3a and Figure 7a,b). This suggests that these two models favored features that have primarily different values in CVD and healthy groups, while the inclusion of NTNG1 served as a supplement to address the remaining variations that had not been explained by the other factors.

Stochastic gradient boosting got the ideal classification of CVD and healthy individuals in both the training set and test set with a superior discrimination of CVD and healthy probability than the linear discriminant analysis and Naïve Bayesian model (Figure 6 and Table 2). The stochastic gradient boosting model is a classification tree-based model that incorporates functionalities that allows iterations over different choices of features and the modeling of residuals at costs until the residual is smaller than a certain threshold. From our modeling practice, the complexity of the stochastic gradient boosting paid off in comparison to the other two tree-based models—namely, the bagged CART model and the random forest model (Figure 6 and Appendix A). The bagged CART model, which does not iterate among different sets of features, suffered from a larger variation of predicted probabilities (Appendix A), suggesting that randomized feature selection did benefit the modeling stability. On the other hand, the random forest model had the ability to produce probability predictions with less variation but failed to deliver a probability above the cutoff in the test set, confirming the extra learning ability made possible by the gradient boosting process (Appendix A). Notably, the stochastic gradient boosting model picked considerably different NGCs as features. The most important three features in this model, SEMA6B, SEMA6D and EPHA2, are not included in the top 10 features in the linear discriminant analysis and Naïve Bayesian models (Figure 7). The most important feature, SEMA6B, was not even significant in the two mean *t*-tests (Figure 3a). Variable importance in the stochastic gradient boosting model took both the importance of a variable in the building of a decision tree and the subsequent modeling of residuals into account. It is likely that SEMA6B performed well in the modeling of the residual, since the univariate prediction power of SEMA6B should be minimal. The modeling of residuals often improves datasets with lower information-to-noise ratios, which is the case in the current dataset.

The importance of a feature in certain models could sometimes inform us with the relevance of the feature to the outcome. In the current setup, the importance of an NGC in the prediction models might suggest the functional importance of NGC in the development of CVD via monocytes. The linear discriminant analysis and Naïve Bayesian model picked up PLXNC1 as the most important feature. PLXNC1 mediates monocyte migration, adhesion and differentiation in response to its ligand SEMA7A [24]. At regions experiencing disturbed blood flow (atheroprone), an increase of SEMA7A in endothelial cells exacerbates inflammation and atherosclerotic plaque size [12]. SEMA6D and EPHA2 were chosen as the second and third most important features in the stochastic gradient boost model. SEMA6D has a role in immunology as being costimulatory molecule-expressed by dendritic cells [25]. The function of SEMA6D in monocytes is not known yet. EPHA2 promotes the adhesion and differentiation of monocytes [26,27]. In an apolipoprotein E knockout murine atherosclerosis model, the knockdown of EPHA2 using adenovirus-carrying short hairpin RNA resulted in the attenuation of atherosclerotic lesion development [28]. However, the amount of contribution from EPHA2 knockdown on monocytes could not be distinguished from that of endothelial cells.

A limitation of this study is that the number of individuals is small in the healthy group, meaning the models were less trained by features of individuals with a healthy phenotype. This also led to some degree of instability of the models in the cross-validation. Another limitation is that the healthy individuals were younger than patients in the CVD group. Although, the ranges of ages in the two groups overlapped, it is to be determined whether the model could discriminate between the two phenotypes when the age range in the healthy group is extended. Future studies should recruit more age-matched individuals. To fully reveal the prediction power of NGC expressions, future works should focus on whether NGC expressions could discriminate between classes with more subtle differences, e.g., between stable and unstable angina. It should also be noted that the pathogenesis of CVD is rather complex. In this article, we focused on the association of NGC expressions to CVD, although multiple pathways are involved, leading to the identification of an incomplete risk gene set. The complex pathogenesis also dampens our ability to draw conclusions on causality, since it is possible that mechanisms that are implicated in CVD also alter the functional states of monocytes, reflected by monocytic gene expressions.

In general, the development of clinical risk prediction models often faces certain hurdles, which could lead to less-defined results. Some studies only examined the univariate prediction value and ignored the combined prediction value of all features [29], while other studies did examine the multivariate prediction value of features but used models that made strong assumptions on the data structure and covariance between features [30,31]. It is also common that studies have accessed the models in a descriptive way, but the external prediction potential is not determined [29,31,32]. In our study, we have incorporated multiple NGCs as features based on both statistical examinations and biological insights. We assessed the performances with cross-validation in the training data and independent prediction in the held-out data, thereby controlling the models on overfitting. Moreover, the models were shown to be not only a descriptive tool to confirm the correlations between NGC expression and CVD outcome but, also, a prediction method that can be applied to new datasets. In addition, we applied one of the complex models, the stochastic gradient boosting, which makes little assumptions on the characteristics of the data structure and returned a better performance. Taken together, this study gave the proof of principle on how machine-learning methods could be applied to the prediction of disease outcomes using the expression of a set of functional genes in circulating cells and allowed us to identify SEMA6B, SEMA6D and EPHA2 as predictive genes for CVD in the current cohort.

## 4. Materials and Methods

### 4.1. Study Population

The study population consists of a subgroup of 369 patients from the CIRCULATING CELLS study cohort [20] (Figure 1). In brief, CIRCULATING CELLS was a prospective multicenter study in which patients scheduled for coronary angiography due to CVD were included. For this subgroup, extensive clinical characteristics were recorded (Table 1), and the transcriptomes of purified circulating CD14+ monocytes were profiled. To minimize the potential influence of the presence of profound acute myocardial ischemia on monocytic gene expression profiles, patients with ST-elevation myocardial infarction (STEMI) were excluded. Additional exclusion criteria were age < 18 years, inability to give informed consent, suspected drug or alcohol abuse, serious concomitant disease, serious recent infectious disease in the last 6 weeks or suspected elevated state of the immune system and noncooperativeness. The CIRCULATING CELLS study was executed in four medical centers in the Netherlands: the Catharina Hospital in Eindhoven, the Leiden University Medical Center, the Maastricht University Medical Center and the University Medical Center Utrecht. The study was approved by the medical ethical committees of the participating centers and conforms to the Declaration of Helsinki, CCMO number NL24071.060.08 approved 25-8-2008. The inclusion period lasted from March 2009 till September 2011. All patients received oral and written information about the objectives of the study and provided written informed consent

### 4.2. Isolation of Peripheral Blood CD14-Positive Monocytes

The full procedures for the isolation of peripheral blood CD14-positive monocytes were described previously [20]. Briefly, 60 mL of EDTA blood was collected from patients via the arterial sheath catheter. Peripheral blood mononuclear cells (PBMCs) were isolated by density gradient centrifugation over Ficollpaque Plus (GE Healthcare, Diegem, the Netherlands). For further purification of monocytes, the PBMC fraction was incubated with magnetic beads coated with anti-CD14 antibodies (BD Biosciences, Breda, the Netherlands), and monocytes were purified with a MACS separation system according to the manufacturer’s instructions (BD Biosciences, Breda, the Netherlands). Cells in CD14-positive fraction were resuspended, lysed in Trizol and aliquoted. The aliquots were stored at -80 °C for RNA isolation.

### 4.3. RNA Isolation and Microarray Analysis

Monocyte samples were shipped to Eurofins Genomics for semiautomated extraction of RNA using RNeasy 96-well plates (Qiagen, Hildencity, Germany). RNA samples were quantified using a Beckman Coulter DTX880 system, and only samples that displayed RIN values > 9 (Agilent Bioanalyzer, Agilent, Santa Clara, CA) were included. Labeled RNA was prepared and used on the array for hybridization. Hybridized chips were scanned by Illumina BeadStation (Illumina, Inc., San Diego, CA, USA). Raw image analysis and signal extraction was performed with Illumina Beadstudio Gene Expression software with default settings (no background subtraction). Data were exported as text files. The gene expression profiling data were integrated and archived using the self-developed software “Circucel” [20]. The expressions of NGC ligands and receptors were extracted along with the phenotypic profiles of the patients. We excluded patient records without the required outcome parameter—in this case, a “confirmed diagnosis”. For NGC expression profiles, there were no missing values. Therefore, data imputation was not necessary.

### 4.4. Statistical Analysis

Univariate correlation of NGC expressions (or other continuous variables) with a categorical variable was tested by 2 mean Student’s *t*-tests. *p*-values were obtained from *t*-statistics. Univariate correlation of NGC expressions (or other continuous variables) with a continuous variable was tested with linear regression. *p-values* were obtained from the *t*-statistic of the coefficient of the variable. Correlation of 2 categorical variables was tested using Pearson’s chi-squared test of the cross-tabulation. *p*-values were obtained from the chi-square statistics. For all the tests, a *p*-value of less than 0.05 was considered significant.

### 4.5. Model Fitting and Assessment of Model Performance

We used R package “caret” and its multiple dependencies (summarized in Table 2) for modeling the predicting power of NGCs to the disease status of patients [33]. Performance statistics for binary classification models were calculated, including accuracy, Cohen’s kappa (*κ*), sensitivity and specificity. The Cohen’s kappa (*κ*) value is given by formula
$$\frac{\left(P\;observed-P\;expected\right)}{\left(1-P\;expected\right)}$$
which indicates the performance gain from the modeling over random guessing (the higher, the better). Since this data set features a smaller number of healthy individuals, we set the models to aim for picking up healthy individuals as events. The sensitivity (true positive prediction) and specificity (true negative prediction) values were also calculated in accord to this principle.

For the external assessment of model performance, data partitions were created to have a training set (90% of the dataset) and a test set (10% of the dataset). This was done using the “createDataPartition” function in the “caret” package to ensure proportional and representative coverage of individuals in both the training and test sets. Data in the training set were used for model building, and the data in the test set were held out in the training process and were used to determine the model performance pseudo-externally by comparing the prediction on the test set with the actual outcome of the test set. The distributions of NGC expressions in the training set and test set were illustrated in Appendix A.

For the internal assessment of model performance and stability, 10-fold cross-validations were done, which means 10% of the training data were kept out of each iteration to evaluate the model generated by the other 90% training data over 10 iterations. Similar performance statistics were calculated at each iteration. The average values and standard deviations of the parameters were summarized to assess the model performance and stability, respectively. Some models were tuned in ranges of tuning parameters to control their complexity and adaptivity (Table 2). Tuning parameters giving the best performance statistics in the cross-validation were chosen as optimal tuning of the model, and a final model was built using these parameters on the complete training data.

## Figures and Tables

**Figure 1 ijms-21-06364-f001:**
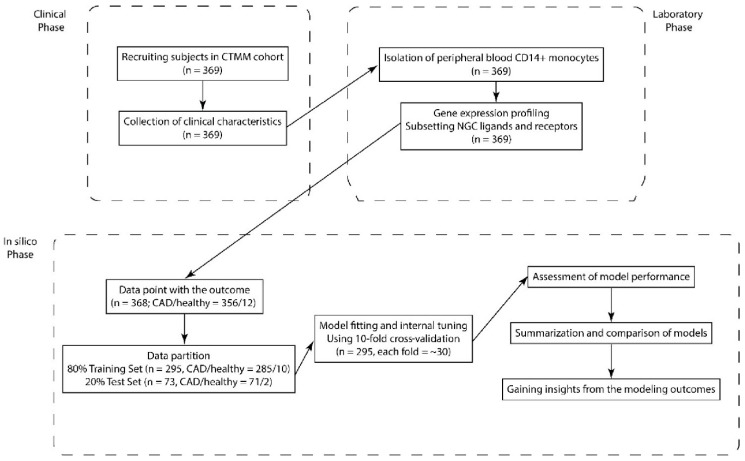
Flowchart of predictive modeling using neuroimmune guidance cues (NGCs). Subjects in the CIRCULATING CELLS cohort were recruited based on inclusion criteria. The clinical characteristics were collected, peripheral blooded CD14-positive monocytes were isolated and their transcriptomes were profiled. The expression of NGCs was subset. The individuals were randomly assigned to the Training Set or Test Set for external assessment of the model performance. Classification models that were built on the Training Set data and model performance were internally assessed with cross-validation. Finally, we made comparisons between the models and gained insights on the choice of model and features.

**Figure 2 ijms-21-06364-f002:**
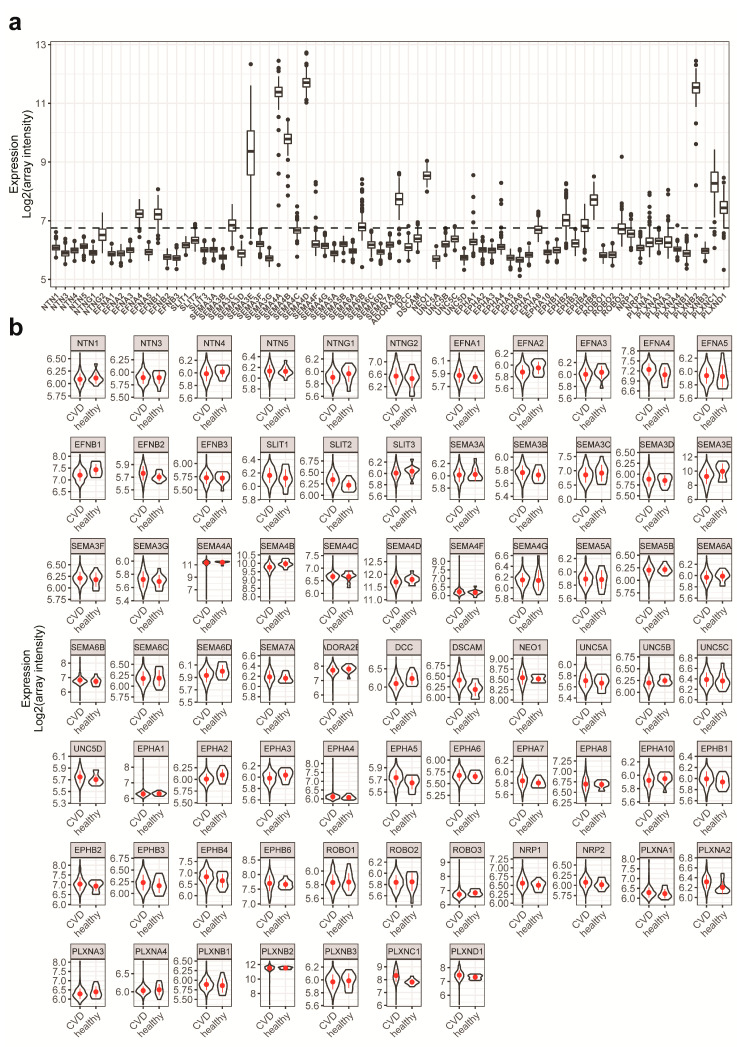
NGC expressions in patients and healthy subjects. (**a**) Box plots with quartiles were created using NGC expressions from all individuals. Baseline signal of the platform (6.75) was indicated with the dashed line. (**b**) Violin plots of all NGC expressions were created for cardiovascular disease (CVD) patients and healthy individuals. The violin shapes represent the density distribution of NGC expressions in the groups. The NGC expressions of CVD patients overlap with those of healthy individuals. Due to the small number of healthy individuals, their NGC expressions were sometimes not normally distributed.

**Figure 3 ijms-21-06364-f003:**
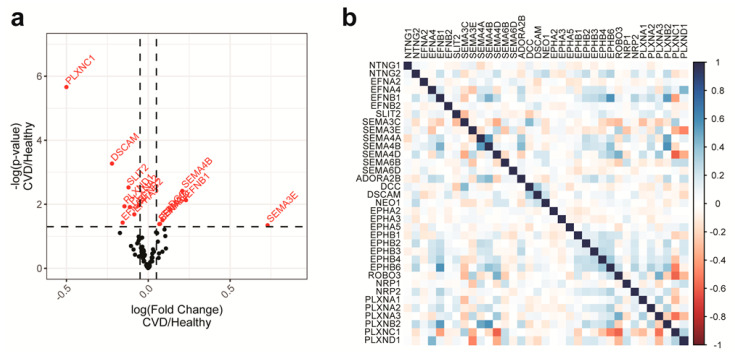
Univariate correlation of NGC expressions with disease phenotypes and pairwise correlations of NGC expressions. (**a**) Volcano plot showing the fold changes and the significance of difference in the 2 mean Student’s *t*-tests between NGC expressions in CVD patients and healthy individuals. NGCs with significant differences (*p* < 0.05) are labeled with gene names and red color. For most NGCs, the univariate correlation determined by the 2 mean Student’s *t*-tests with the outcomes is minimal. (**b**) Pairwise covariances of NGC expressions were illustrated in the heatmap to examine the between-feature correlations. Blue color indicates a positive correlation, whereas red color indicates a negative correlation. There are no absolute pairwise covariances higher than 0.75, the common cutoff for highly correlated features.

**Figure 4 ijms-21-06364-f004:**
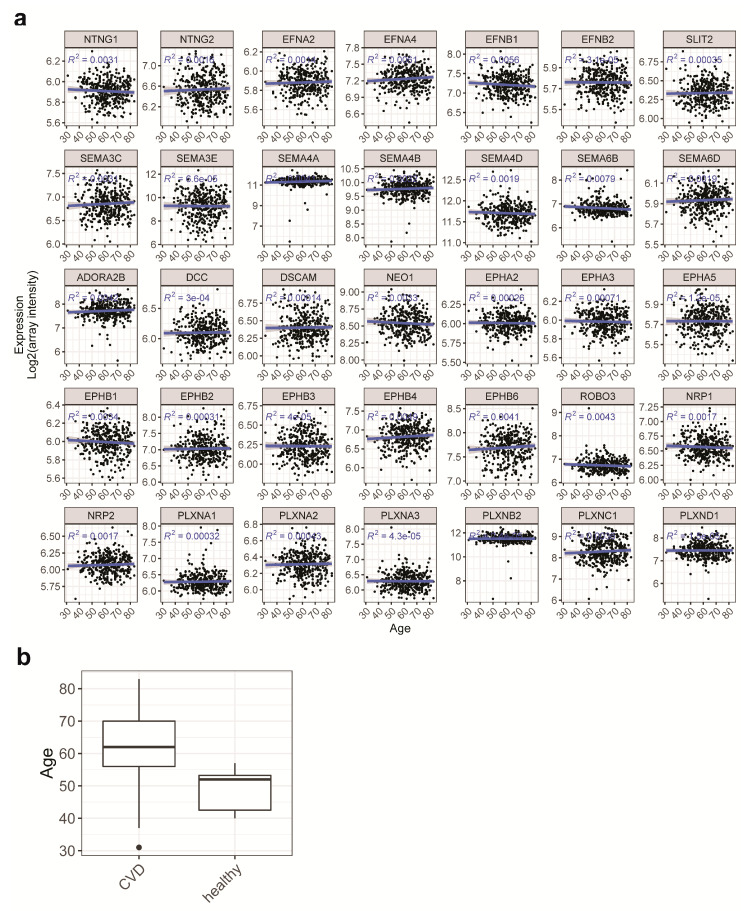
Influence of age as a potential confounding factor. (**a**) Scatter plots of expressions of selected NGCs in relation to age were made to show the influence of age on NGC expressions. Linear regressions were done with age being the dependent variable providing the fitted trend line (blue line), the 95% confidence interval of the trend (gray area) and the regression R-squared. Correlations of NGC expressions and age are minimal, as indicated by the R-squared values. (**b**) Boxplot of age distribution in CVD patients and healthy individuals. Young age dominates in the healthy individuals.

**Figure 5 ijms-21-06364-f005:**
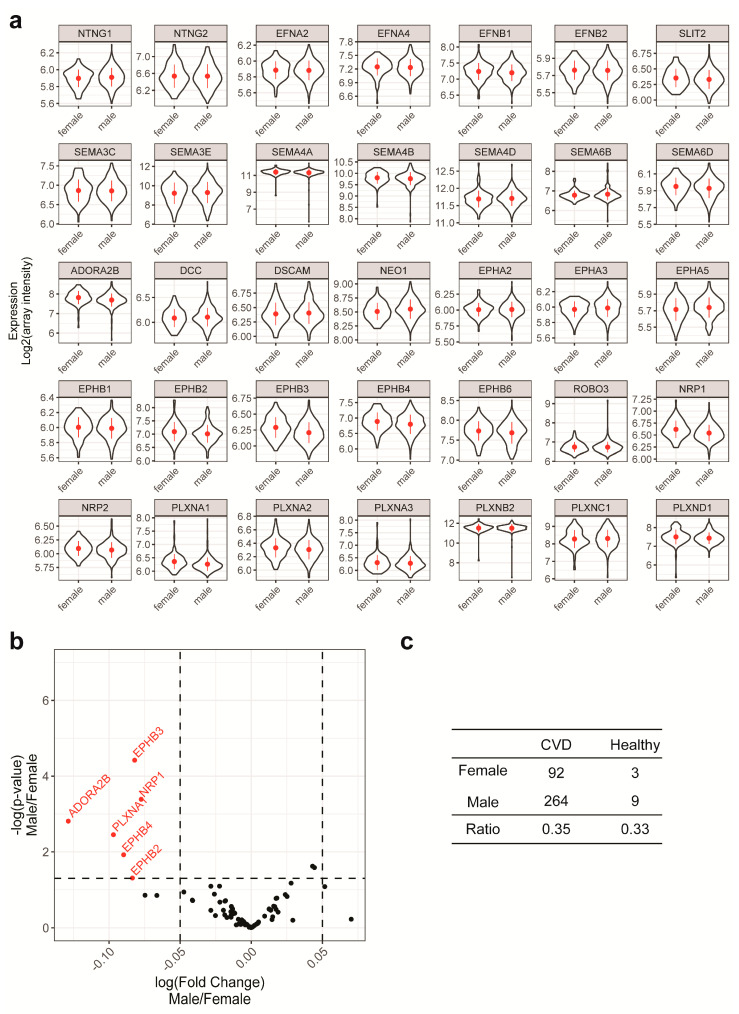
Inspection of the influence of sex as a potential confounding factor. (**a**) Violin plots to show the distribution of expressions of selected NGCs in males and females. To be noticed is that distributions of selected NGC expressions are similar between sexes. (**b**) A volcano plot was made showing the fold change and the significance of differences in a 2 mean Student’s *t*-tests between NGC expressions in males and females. NGCs with significant differences (*p* < 0.05) were labeled with gene names and red color. (**c**) Contingency table showing that the ratio of sexes is not biased in relation to the outcomes.

**Figure 6 ijms-21-06364-f006:**
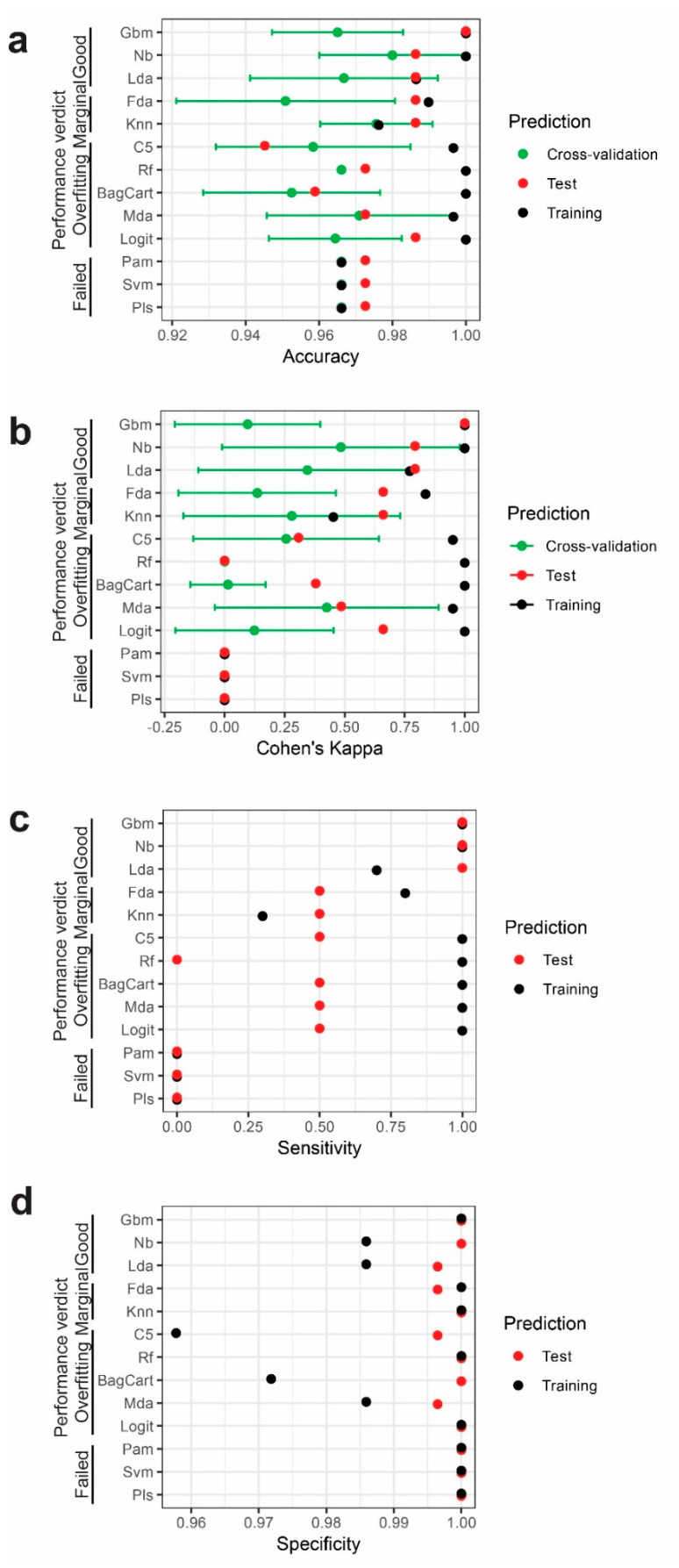
Model performance. (**a**–**d**) Model performance metrics—namely, accuracy (**a**), Cohen’s Kappa (**b**), sensitivity (**c**) and specificity (**d**)—were illustrated in cross-validations (green dots and error bars), Training Set (black dots) and Test Set (red dots) for all models. Models are categorized into 4 groups based on their performances.

**Figure 7 ijms-21-06364-f007:**
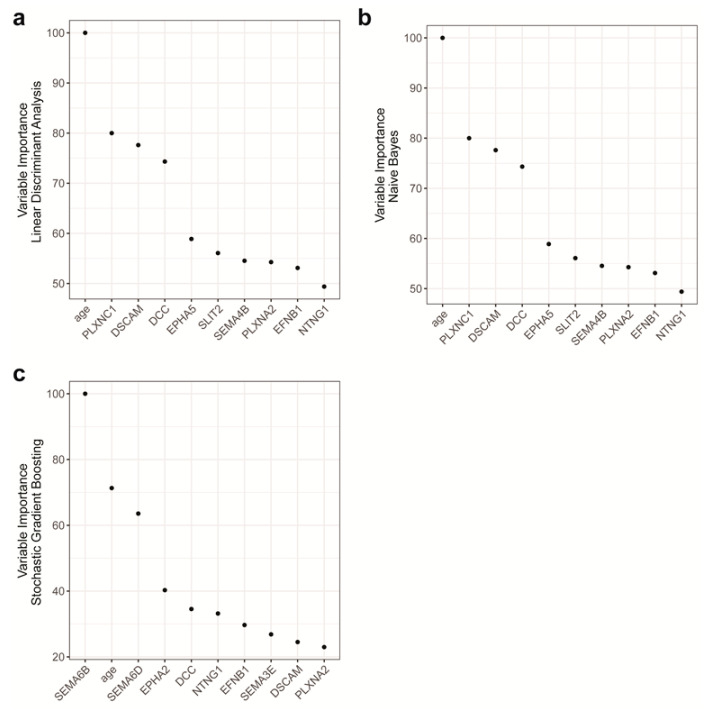
Variable importance of the models. (**a**–**c**) Variable importance measured in the linear discriminant analysis (**a**), Naïve Bayesian model (**b**) and stochastic gradient boost model (**c**). The importance of the most important feature was scaled to 100.

**Table 1 ijms-21-06364-t001:** Clinical characteristics of the CIRCULATING CELLS cohort.

Clinical Characteristics	All	CVD	Healthy
*Demographic data*			
Number (male/female)	368 (273/95)	356 (264/92)	12 (9/3)
Age	61.8 (±10.4)	62.2 (±10.3)	49.2 (±6.3)
BMI	27.3 (±4.3)	27.4 (±4.3)	23.7 (±2.3)
*Coronary risk factors*			
Hypertension	231 (63%)	231 (65%)	0 (0%)
Hypercholesterolemia	70 (19%)	65 (18%)	5 (42%)
Diabetes	77 (21%)	77 (22%)	0 (0%)
Current smoker	73 (20%)	73 (21%)	0 (0%)
Family MI history	141 (39%)	137 (39%)	4 (33%)
Previous MI	112 (30%)	112 (31%)	0 (0%)
Positive family history	157 (43%)	152 (43%)	5 (42%)
*Therapeutic decision*			
PTCA	130 (35%)	130 (37%)	0 (0%)
CABG	32 (9%)	32 (9%)	0 (0%)
*NYHA Classification*			
NYHA Class I	248 (67%)	236 (66%)	12 (1%)
NYHA Class II	78 (21%)	78 (22%)	0 (0%)
NYHA Class III	26 (7%)	26 (7%)	0 (0%)
NYHA Class IV	16 (4%)	16 (4%)	0 (0%)
*Current medication*			
β-blocker	228 (69%)	228 (72%)	0 (0%)
Ca-antagonist	95 (29%)	95 (30%)	0 (0%)
Aspirin	260 (79%)	260 (82%)	0 (0%)
Vitamin K antagonist	29 (9%)	29 (9%)	0 (0%)
Low molecular weight heparin	10 (3%)	10 (3%)	0 (0%)
ADP receptor blocker	168 (51%)	168 (53%)	0 (0%)
ACE inhibitor	116 (35%)	116 (36%)	0 (0%)
ATII receptor blocker	71 (22%)	71 (22%)	0 (0%)
Diuretic	76 (23%)	76 (24%)	0 (0%)
Statins	252 (77%)	252 (79%)	0 (0%)

Values are N ± SD or N (%). Abbreviations: BMI—body mass index, MI—myocardial infarction, PTCA—percutaneous transluminal coronary angioplasty, CABG—coronary artery bypass graft, NYHA—New York Heart Association, and CVD—cardiovascular disease.

**Table 2 ijms-21-06364-t002:** Summarization of model names, types and tuning parameters.

Model Name	Abbreviation	Type	Best Tuning Parameter (Tuning Range)	R Package Dependency
Boosted Logistic Regression	Logit	Linear	Number of iterations = 41 (11, 101)	“caTools”
Linear Discriminant Analysis	Lda	Linear	NA	“MASS”
Partial Least Squares	Pls	Linear	Number of components = 1 (1, 10)	“pls”
Support Vector Machines	Svm	Nonlinear	Cost = 0.25 (2 × 10^−2^, 2 × 10^−7^)	“kernlab”
Nearest Shrunken Centroids	Pam	Linear	Threshold = 0 (0, 25)	“pamr”
Mixture Discriminant Analysis	Mda	Nonlinear	Subclasses = 11 (2, 16)	“mda”
Flexible Discriminant Analysis	Fda	Nonlinear	Degree = 4 (1, 5); Number of pruning = 5 (2, 5)	“earth”, “mda”
k-Nearest Neighbors	Knn	Nonlinear	Number of neighbors = 5 (5, 23)	“class”
Naive Bayesian	Nb	Nonlinear	Laplace correction = 1 (1, 3); Kernal function = FALSE (FALSE, TRUE); Bandwidth Adjustment = 1 (1, 3)	“naivebayes”
Bagged CART	BagCart	Tree/Rule-based	NA	“ipred”, “plyr”, “e1071”
Random Forest	Rf	Tree/Rule-based	Number of random parameters = 2 (2, 37)	“randomForest”
Stochastic Gradient Boosting	Gbm	Tree/Rule-based	Interaction depth = 1 (1, 7); Number of trees = 450 (100, 1000); Shrinkage = 0.1 (0.01, 0.1); Min terminal node size = 5 (5, 7)	“gbm”, “plyr”
Single C5.0 Tree	C5	Tree/Rule-based	NA	“C50”, “plyr”

**Table 3 ijms-21-06364-t003:** Confusion matrices of the linear discriminant analysis, Naive Bayesian and stochastic gradient boosting models.

	Training	Reference		Test	Reference	
Model	Prediction	CVD	Healthy	Prediction	CVD	Healthy
Linear Discriminant	CVD	284	3	CVD	70	0
Analysis	Healthy	1	7	Healthy	1	2
Naive Bayesian	CVD	285	0	CVD	70	0
	Healthy	0	10	Healthy	1	2
Stochastic Gradient Boosting	CVD	285	0	CVD	71	0
	Healthy	0	10	Healthy	0	2

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
