# Peer review of "Prediction Power on Cardiovascular Disease of Neuroimmune Guidance Cues Expression by Peripheral Blood Monocytes Determined by Machine-Learning Methods"

_ijms, 2020, doi:10.3390/ijms21176364_

Round 1
Reviewer 1 Report
In the submitted manuscript, Zhang and colleagues test the utility of various statistical tools to predict cardiovascular risk based on the expression levels of neuroimmune guidance cues (NGC) in monocytes. While the study does not bring any novel insights into the involvement of NGC in the pathogenesis of coronary disease, it is of interest from the methodological point of view, as it assesses on a practical example the utility of various risk modelling approaches. The methodology of the study seems to be sound and the manuscript is professionally written. My major concern is the exceedingly small control group, which encompasses only ten individuals. The authors must convincingly explain, why in their view the tiny control group does not affect the results of the study. They should also comment, why the control group used in the study is very significantly younger than the cardiovascular group.
Minor comment
The legibility of figures should be improved – at present some of them can be effectively read only with a magnifying glass.
Reviewer 2 Report
The study titled “Prediction power on cardiovascular disease of neuroimmune guidance cues expression by peripheral blood monocytes determined by machine learning methods” by Zhang et. al., is impressive. The study has been conducted properly, results are analyzed properly, and findings are important for the development of a predictive method for early detection of CVD. In this study, authors have used Neuroimmune Guidance Cues (NGC) expression of peripheral blood monocytes for the prediction of CVD. They also performed cross- validation, and validation using a pseudo-external test set with conventional confounding factors controlled. They tested various models and found that Naïve Bayesian model and stochastic gradient model had satisfactory discrimination in both training test and test set. The stochastic gradient model with residual modeling mechanism was able to achieve 100% accuracy. Thus, authors have demonstrated a proof of concept that a small set of functional genes, NGCs, is of enough prediction power for classification of CVD patients and healthy individuals.
The manuscript is well written and has the necessary statistical analyses. I recommend the acceptance of the manuscript.
Thank you.
Author Response
We would like to thank the reviewer for his/her thoughtful comments regarding our study. We are pleased that the reviewer thought the manuscript was impressive, conducted properly, and well-written.